# *f*HER2, PR, ER, Ki-67 and Cytokeratin 5/6 Expression in Benign Feline Mammary Lesions

**DOI:** 10.3390/ani12131599

**Published:** 2022-06-21

**Authors:** Maria Soares, Assunção N. Correia, Mariana R. Batista, Jorge Correia, Fernando Ferreira

**Affiliations:** 1CIISA—Centre for Interdisciplinary Research in Animal Health, Faculty of Veterinary Medicine, University of Lisbon, 1300-477 Lisbon, Portugal; mariajosoares@gmail.com (M.S.); marianabatista@fmv.ulisboa.pt (M.R.B.); jcorreia@fmv.ulisboa.pt (J.C.); 2Associate Laboratory for Animal and Veterinary Sciences (AL4AnimalS), 1300-477 Lisbon, Portugal; 3IUEM—Instituto Universitário Egas Moniz, Egas Moniz—Cooperativas de Ensino Superior, Campus Universitário, Quinta da Granja, Monte da Caparica, 2829-511 Caparica, Portugal; 4Basic Sciences Academic Division, FMV-ULHT—Faculdade de Medicina Veterinária, Universidade Lusófona de Humanidades e Tecnologia, Campo Grande 376, 1749-024 Lisboa, Portugal; acorreia@gmail.com

**Keywords:** premalignant lesions, feline oncology, estrogen receptor, progesterone receptor, HER2, Ki-67, cytokeratin 5/6

## Abstract

**Simple Summary:**

Feline mammary neoplasias are highly prevalent in domestic cats and present many similarities to their human counterparts. Since information about benign feline mammary lesions is still scarce and often controversial, studies using a wider panel of oncological biomarkers are necessary to understand their potential contribution to malignant lesions. This study analyzed 47 benign lesions from 27 queens, regarding the expression of estrogen and progesterone receptors (ER and PR, respectively), *f*HER2 protein and two malignancy indicators (Ki-67 and CK 5/6). Our results showed that most of the lesions were ER positive (91.5%), PR negative (63.8%), *f*HER2 negative (64.4%), Ck 5/6 negative (76.6%) and with a low Ki-67 index (78.7%). Additionally, significant correlations were found between younger ages and ER positivity and between larger lesions and negative PR status. Our results highlight the importance of estrogen receptors in the development of benign feline mammary lesions, further contributing to the development of preventive and monitoring strategies in feline mammary oncology.

**Abstract:**

Biomarkers are essential in the characterization of neoplastic lesions and aid not only in the classification of the nature of the lesions, but also in the understanding of their ontogeny, development and prognosis. In cats, while mammary carcinomas are increasingly being characterized, information on their benign lesions is still scarce. Indeed, a better characterization of benign lesions could have an important role in unravelling mammary oncogenesis, similar to that in human breast cancer. Thus, in this study, the expression of five markers was analyzed in 47 benign mammary lesions (hyperplasia, dysplasia and benign tumors) collected from 27 queens. Dysplastic and hyperplastic lesions were the most common (41/47, 81.7%). Most of the lesions were classified as ER positive (43/47, 91.5%), PR negative (30/47, 63.8%), *f*HER2 negative (29/47, 64.4%), CK 5/6 negative (36/47, 76.6%) and with a low Ki-67 index (37/47, 78.7%). Statistical analysis revealed a correlation between younger ages and ER positivity (*p* = 0.013) and between larger lesions and negative PR status (*p* = 0.038). These results reinforce the importance of evaluating the expression of the ER status, prevalent in benign lesions, as a putative precursor in cancer progression.

## 1. Introduction

Cancer develops through a complex multistage process (histogenetic and molecular modifications), where preneoplastic and precancerous lesions can evolve into malignant tumors [1,2]. Preneoplastic lesions, such as focal hyperplasia or dysplasia can differentiate into benign tumors, which do not usually invade surrounding healthy tissues and maintain a well-differentiated histological pattern. Precancerous lesions, which include adenomas, can evolve and eventually originate malignant neoplasias such as carcinomas [2]. The sequence of events leading up to the development of malignant lesions depends on a multitude of factors and genetic alterations that are widely accepted as major players in this process. As neoplasia develops, normal cells progressively acquire aberrant competences such as the capacity to evade apoptosis, anti-growth-signal insensitivity or growth signal autonomy, which ultimately leads to their full transformation into invasive and malignant tumor cells [3,4,5].

The early identification of neoplasia development events is of paramount importance in both human and veterinary oncology, as it allows for more precocious and efficient treatments and improved prognostics. As a consequence, several biomarkers have been studied in preneoplastic, precancerous and neoplastic tumors. In addition to their inherent importance in understanding the biology and predicting the behavior of tumor cells, these biomarkers could also be used as targets for therapeutic agents aiming to block or reverse tumor progression, ideally at the early stages of the disease [6].

Breast cancer is the most common cancer and is still the leading cause of cancer-related deaths in women [7]. Although mammary tumors have been described as the third most common cancer in cats, a recent study indicates that their occurrence could be more common, being the most frequently diagnosed feline cancer in recent years [8,9,10,11]. Furthermore, feline mammary tumors typically present very aggressive behavior and usually metastasize, thus causing animal death [8]. Another important feature of feline mammary cancer is its similarity to its human counterpart, human breast cancer (HBC), which makes the feline species a potential animal model for comparative oncology, especially regarding HER2 and triple negative mammary tumors [12,13].

Several studies have implicated a multitude of genomic and proteomic changes occurring in the early stages of the disease in humans [14,15,16,17]. Particularly, lesion expression of the estrogen receptor (ER), progesterone receptor (PR), HER2 and Ki-67 have been identified as promising biomarkers to monitor the progression of invasive breast cancers in humans [16,17,18]. Indeed, hormone receptors (ER and PR) are involved in the regulation of the physiological growth and differentiation of breast epithelium promoted by estrogen [19,20,21]. Estrogen is considered one of the most potent carcinogenic factors in women and its receptor has been investigated in precursor lesions, with ER overexpression being described in hyperplastic lesions [19,22,23]. In cats, some studies have evaluated ER expression in mammary dysplastic lesions, showing that the receptor’s expression is increased in normal and dysplastic tissue when compared to malignant lesions [24,25]. Additionally, some studies show that benign feline mammary lesions have a higher expression of PR when compared to malignant lesions [20,26].

HER2/neu is a protooncogene that encodes a transmembrane tyrosine kinase receptor belonging to the epidermal growth factor receptors’ family. The amplification and/or overexpression of HER2 has a relevant prognostic value, as it is associated with shorter overall survival and disease-free survival [27]. Although HER2 overexpressing tumors are more aggressive, they are also sensitive to HER2-targeted therapies, such as monoclonal antibodies or tyrosine kinase inhibitors, rendering them more susceptible to currently available treatments [16,28]. In HBC, HER2 amplification or overexpression can occur in the early stages of the disease, where the activation of the *Her2* gene increases from benign lesions to invasive tumors. Thus, some authors point out that HER2 is involved in breast oncogenesis and is an indicator of tumor invasiveness [16,29]. In feline mammary tumors, HER2 expression has been extensively investigated, and was found to be overexpressed in 30 to 50% of mammary carcinomas [30,31,32,33,34].

Cellular proliferation is an integrant part of tumor development and several studies have investigated the expression of proliferation markers in preneoplastic and precancerous lesions [35,36,37]. Among these markers, Ki-67 has been associated with the worst prognosis in both HBC and feline mammary carcinomas [17,38,39]. Likewise, patients with mammary tumors that express basal-type cytokeratins (CK), such as 5 and 6, have been shown to have a poorer prognosis [40].

Considering the scarce information regarding the presence and importance of ER, PR, *f*HER2, Ki-67 and CK5/6 proteins in dysplastic lesions and benign mammary tumors, the aim of this study was to provide such a characterization and investigate its putative association to clinicopathological parameters.

## 2. Materials and Methods

### 2.1. Animal Population

Animals enrolled in this study underwent surgical treatment to extirpate benign mammary lesions (benign tumors and benign non-neoplastic lesions) at 6 veterinary clinics in the Lisbon area, including the Teaching Hospital of the Faculty of Veterinary Medicine, University of Lisbon (FMV-ULisboa). Sample collection was performed from 27 queens and all owners gave their consent to tissue sample collection and for the use of the animal’s clinical data. Enrollment in the study did not influence cases’ clinical management, nor further interfere with the animals’ well-being. Data regarding age, breed, and reproductive status were recorded for each animal. Data regarding previous contraceptive administration were available and recorded for all animals, except in one case, where the owner could not provide the information.

### 2.2. Mammary Tissue Collection and Histopathological Classification

Mammary samples were collected during surgery under general anesthesia and were fixed in 10% buffered formalin, for 24 to 48 h, warranting optimal fixation conditions for immunostaining and routine diagnosis [41,42,43]. Samples were then processed using routine histological procedures and stained with hematoxylin and eosin. All lesions were classified by a trained veterinarian pathologist using the World Health Organization (WHO) classification system, as adapted by Zappulli et al., 2019 [44]. For the present study only the benign lesions (benign tumors and benign non-neoplastic lesions) were considered and used for the immunohistochemistry analysis.

In addition to histological classification, all lesions were measured after resection and after fixation, and the larger diameter was recorded. Coexisting malignant tumors detected in other resected mammary glands were also noted for the present study.

### 2.3. Immunohistochemistry Analysis

Immunohistochemistry (IHC) detection of *f*HER2, Ki-67, PR, ER and CK 5/6 was performed on tissue microarrays, using our laboratory’s routine protocols, as previously published [12]. Briefly, a representative area of each lesion (0.6 cm in diameter) was selected and used to prepare five serial 3 µm sections on microscope glass slides (SuperFrost Plus microscope slides, Thermo Scientific, Rockford, IL, USA). Slides were dried at room temperature, heated at 60 °C in a dry incubator for 1 h and finally deparaffinized. For ER, *f*HER2 and Ki-67 immunodetection, antigen retrieval was performed by boiling samples in a pressure cooker for 2 min at 2 atm in a sodium citrate buffer solution (0.01 M NaCH_3_COO, pH 6.0). For PR immunodetection, antigen retrieval was achieved with sodium citrate buffer in a water bath at 95 °C for 60 min and for CK 5/6, samples were microwaved at 900 W for 15 min in Tris-EDTA buffer (pH 9.0). After antigen retrieval, samples were treated with peroxidase block (peroxide-block solution, Max Polymer Detection System, Leica Biosystems, Wetzlar, Germany) for 10 min, followed by unspecific protein block (protein-block solution, Max Polymer Detection System, Leica Biosystems) for 10 min. Following blocking treatments, slides were incubated with the following primary antibodies: anti-ER monoclonal antibody (clone 6F11, 1:100 dilution, overnight at 4 °C; Thermo Scientific), anti-PR monoclonal antibody (clone 1E2, ready to use, overnight at 4 °C; Ventana, Tucson, AZ, USA), anti-HER2 monoclonal antibody (clone CB11, 1:200 dilution, overnight at 4 °C; Invitrogen, Carlsbad, CA, USA), anti-Ki-67 polyclonal antibody (PA5-19462, 1:500 dilution, 1 h at room temperature, Thermo Scientific) and anti-CK 5/6 monoclonal antibody (clone D5/16B4, ready to use, overnight at 4 °C; Ventana). Finally, the post-primary reagent (Max Polymer Detection System, Leica Biosystems) and the Novolink polymer (Max Polymer Detection System) were applied for 30 min each. Afterwards, sections were stained with DAB (3,3′-diaminobenzidin-tetrahydrochlorid) chromogen (Dako, Glostrup, Denmark) for 5 min and counterstained with Mayer’s hematoxylin (Merck, Rahway, NJ, USA). Negative and positive controls were used for each immunostaining protocol: feline uterus was used as positive control for ER and PR, human cell lines (SKBR3) and feline mammary samples known to be positive for HER2 were used for *f*HER2, feline tonsil was used as positive control for Ki-67, and normal feline oral mucosa was used as positive control for CK5/6. 

Protein expression score was evaluated by two independent observers and discordant interpretations were debated using a multiviewer microscope. Immunostaining interpretation of ER, PR and HER2 slides was carried out following ASCO guidelines, as described in Table 1 [45,46]. HER2 status was considered positive when scored 2+ or 3+, as previously described in feline studies [39,47]. Allred score [48] was also calculated for PR and ER expression, for which the classification criteria are explained in Table 2. Ki-67 index was calculated by dividing the number of positive nuclear stained lesions cells (hyperplastic/dysplastic/tumor cells) by the total number of observed lesions cells (at least 1000 cells), and samples were considered highly proliferative with at least 14% positively stained cells [39]. For CK5/6 evaluation, tissue samples were considered positive when more than 1% of lesions cells presented cytoplasmic and/or cytoplasmic membrane staining [49]. 

### 2.4. Statistical Analysis

Statistical analysis was performed using SPSS (Statistical Package for the Social Sciences, version 26.0, IBM, New York, NY, USA) software. Two-tailed *p* values lower than 0.05 were considered significant. Samples were found to not follow a normal distribution when subjected to the Shapiro–Wilk test. Consequently, associations between the different biomarkers (ER, PR, HER2, Ki-67 and CK5/6 status) and between biomarker’s staining and clinicopathologic features (age, breed, reproductive status, contraceptive administration, histopathological classification, size of the lesion and the presence of coexisting malignant tumors) were evaluated using Fisher’s exact test, Mann–Whitney and Kruskal–Wallis tests. Possible associations between the different biomarkers were also evaluated. Spearman correlation was used to test correlations between continuous variables (animal age, lesion size and Ki-67 index, PR and ER Allred score and CK 5/6 staining).

## 3. Results

### 3.1. Clinicopathological Features

The clinicopathological findings from the 27 animals are summarized in Table 3. The median age of the animals presenting with mammary dysplastic and hyperplastic lesions and benign tumors in this study was 9.6 years (SEM ±3 years) and most animals were European Shorthair (74.1%, 20/27). Most animals (70.3%, 19/27) were not spayed at the moment of the diagnosis and more than half (57.7%, 15/26) had been subjected to progestin therapy as contraceptive. When statistical analysis was performed, no statistical correlation was found between spayed status or progestin administration and the number of lesions presented by the female cat at the time of the diagnosis (including the malignant lesions, when they were present).

Lesions were collected from animals undergoing their first and second (and in one case their third) surgery. Notably, 16 of the 27 animals enrolled in this study (59.3%), presented concomitant malignant tumors in other mammary glands, although no significant correlation was found between the presence of malignant tumors at any time and the number of surgical interventions performed. 

Forty-seven benign lesions (benign tumors, dysplastic and hyperplastic lesions) were collected from 27 animals, since 12 female cats presented more than one benign mass. However, only five cats showed different benign lesions in different mammary glands, while the more usual was the presence of the same lesion (e.g., duct ectasia) in several mammary glands.

From the forty-seven lesions evaluated in this study, 41 were classified as dysplastic or hyperplasic lesions (87.2%, 41/47), and 6 as benign mammary neoplasias (18.2%, 6/47) (Table 4). Duct ectasia was the most common lesion, representing nearly half of all benign lesions. The third most frequent was the simple adenoma, a benign tumor, corresponding to all diagnosed benign mammary neoplasias. 

The mean size of the lesions (benign tumors, dysplastic and hyperplastic lesions) at the time of surgery was 1.76 cm (SEM ± 0.35), ranging between 0.1 and 10 cm. No significant differences were found between overall hyperplasias and dysplasias (1.77 ± 0.37 cm) and benign neoplasias (1.75 ± 1.08 cm). However, fibroadenomatous changes present a significantly higher size than duct ectasias and epitheliosis (*p* = 0.029 and *p* = 0.02, respectively), although not different from adenosis (*p* = 0.17).

### 3.2. Immunohistochemical Analysis 

The results of the immunohistochemical analysis are summarized in Table 5, Figure 1 and Figure 2. Most of the benign non-neoplastic lesions (hyperplastic and dysplastic lesions) presented a positive ER status (95.1%), a negative PR and *f*HER2 status (61% and 64.1%, respectively), a low Ki-67 index (70%) and absence of CK5/6 expression (70%).

Regarding the Allred score classification, the median score was 6.34 (SEM ±0.33) for the ER, with the higher score (8/8) as the more common result (38.3% of the lesions), and most of the lesions having a score between 6 to 8 (80.8% of the lesions). For the PR, the Allred score presented different results, with a mean score of 1.74 (SEM ± 0.33). In fact, 57.4% of the lesions do not show any immunostaining and just one lesion (2.1%) presented the maximum score (8/8).

Immunohistochemical analysis revealed that benign tumors had a similar expression of ER, PR, *f*HER2, Ki-67 and CK5/6 to the aforementioned benign non-neoplastic lesions, with 67% of lesions showing ER positive status, with an *f*HER2 and a CK 5/6 negative status. In addition, PR expression was considered negative and Ki-67 was considered low, in most of the observed adenomas (83.3% for both markers). Four of the studied lesions (9%) did not express either ER or PR, but all the others expressed at least one of these receptors.

Finally, several correlations were found between the clinicopathological features and the expression status of the different markers. The ER Allred score was negatively correlated (*p* = 0.013, r = −0.46) to animal age, with higher scores (more ER staining) associated with younger animals. Similarly, *f*HER2 positive status was also shown to be associated with younger animals, although this correlation was not statistically significant (*p* = 0.064). 

In relation to the size of the lesions, a significant association was found between the larger lesions and negative PR status (*p* = 0.023). This association was present (*p* = 0.038) even when the ductal ectasias were removed from the analysis, which was decided because the increase in the size of these lesions could be more dependent on the dilation of the duct than on the cellular proliferation.

When statistical analysis was performed to investigate possible associations between the different biomarkers, no significant result was found.

### 3.3. Reproductive Status and Hormone Exposition and Expression of the Hormone Receptors

Most of the cats were not spayed at the time of the diagnosis (19/27, 70.4%) and presented different results for ER and PR expression. In fact, 89.5% of the queens (intact female cats) presented lesions with a positive staining for ER (17/19), and negative staining for PR (14/19, 73.7%). Moreover, 57.9% of this subpopulation were subjected to progestin therapy (11/19). 

Regarding the spayed animals (8/27, 29.6%), almost all of them demonstrated positive lesions for ER (7/8, 87.5%), but more lesions were PR positive (4/8, 50%), when comparing with the spayed group. Half of them were also exposed to contraceptive therapy (4/8, 50%).

The animals that were exposed to progestin therapy (15/26, 57.7%) developed lesions that were mainly ER positive (12/15, 80%) and PR negative (11/15, 73.3%), similar to the results obtained for the lesions of the intact female cats.

## 4. Discussion

The characterization of preneoplastic and precancerous lesions is fundamental for a better understanding of the oncogenesis and the improvement of preventive strategies that avoid the evolution of such lesions into lethal malignant and invasive tumors. In feline oncology, studies devoted to the characterization of mammary lesions are scarce, usually focusing on malignant lesions, and are restricted to the one or two biomarkers at a time [13,20,24,26,34]. In fact, there is only one study which characterizes the expression of multiple biomarkers [25], namely ER, PR, *f*HER2. However, this characterization was made mainly in malignant lesions, leaving an important hiatus in information regarding benign neoplasias. Thus, the need for further work that completes and validates the scarce information currently available is currently pertinent. The current study contributes to the advancement of the current knowledge in this area, since it has identified associations not yet reported in the literature.

Several studies on human breast cancer, have investigated possible genetic alterations in benign lesions and identify ER, Ki-67 and HER2 expression as useful biomarkers in monitoring disease progression [1,15,18,50,51].

Although Caliari et al. (2014) evaluated 79 lesions (73 non-neoplastic benign tumors and 6 benign tumors) [25], most of the studies in the literature use a small sample of benign lesions (4 to 20 samples) [20,24,34]. The biggest of these studies found that the most common hyperplastic and dysplastic lesions were the lobular hyperplasia and the ductal ectasia [25]. The characterization of the benign feline mammary lesions in our study is in accordance with what was previously observed in domestic carnivores [20,24,25], once 87% of all analyzed mammary masses were classified as dysplastic or hyperplasic, which are potential preneoplastic lesions. Among these, ductal ectasia was the most common lesion. Regarding the benign neoplasias, only adenomas were diagnosed in our sample collection. 

The majority of currently available studies reports a positive ER status in most benign lesions [24,25], while there are other studies with contrasting information [20], further exemplifying the importance of more studies using larger samples. Our results support the findings of de las Mulas et al. [24] and Caliari et al. [25], since our sample had a higher percentage of positive ER lesions (91.5%). While most cats were not spayed at the time of diagnosis (19/27, 70.4%), and most lesions were classified as ER-positive (17/19), only 26.3% of the intact queens presented a PR-positive status in their lesions (5/19), implying that the dependence of sexual hormones was mainly on oestrogens and not progesterone. Considering the oestrous cycle of queens in which ovulation is induced, and therefore progesterone is scarce if the animal does not mate, these lesions should be able to continue their development in normally cycling queens. Despite this, our study did not find any statistical association between ER or PR status (or any other biomarker) and ovariectomy (and consequent removal of naturally occurring oestrogens) or contraceptive administration.

In HBC, oestrogen has a recognized role in cancer progression. Precursor lesions present a higher expression of oestrogen receptors, which are involved in the transition from normal cells to hyperplastic lesions and then to carcinoma in situ [19,51], strongly suggesting that ER status assessment is critical for the clinical management of premalignant breast disease, being one of the deciding factors when choosing chemoprevention, such as antihormonal drugs [52].

ER positive staining was significantly correlated with younger ages, demonstrating that these less aggressive lesions (non-neoplastic benign tumors and benign tumors), express oestrogen receptors and are potentially dependent on the hormone for development. 

Contrasting with the ER, the PR status was negative in most lesions (n = 30/47, 63.8%). The few studies in feline mammary lesions investigating the presence of this hormone receptor revealed different percentages of positivity, with most of the dysplastic lesions presenting an overexpression [20,26]. However, these studies were performed on very small samples (n= 9, and n = 21) and therefore the advantages of further complementing them are self-evident. In HBC, PR expression is less characterized and information regarding the role of progestins (which activate PR) is still controversial, as it is still not clear whether they stimulate or inhibit tumor growth [22,53]. They are usually used as a surrogate marker of ER functionality, with some studies also pointing to PR overexpression in premalignant lesions [51]. Our results found an association between PR and lesion size, as larger lesions are associated with a negative PR status. The pathophysiological meaning of this finding is still unclear, but it is known that hormone receptor negative status is associated with more aggressive behavior in malignant lesions [54]. In fact, PR and ER negative statuses are associated with higher microvascular density (MVD) and VEGF expression in HBC and in FMC, which are recognized angiogenic markers, and thus facilitate tumoral growth [55,56,57]. 

HER2 is a well-recognized proto-oncogene in HBC, with prognostic and predictive value, and several target therapies against HER2 have improved overall survival in women with breast cancer, showing also promising results on feline cell line studies [23,33,45]. However, its role in precancerous lesions is not well defined, and while some studies show that most benign lesions are HER2 negative, others associate its overexpression in premalignant breast lesions with increased proliferation [23,52]. Some studies investigating HER2′s expression in benign feline mammary lesions found the prevalence to be low (0 to 25%) [13,34,48]. However, our study found that 35.6% of the analyzed lesions had an HER2-positive status. While less than half of the lesions expressed HER2, this is a higher percentage than expected from the literature, reaffirming the necessity to revaluate this receptor’s importance in benign feline lesions, and further studies are needed to confirm its definite role in potential precancerous and preneoplastic lesions.

Another well-known biomarker is Ki-67, which is expressed in cells undergoing mitosis and is therefore, a reliable marker for cellular proliferation. Although proliferative activity is increased in the progression of carcinogenesis, studies point to lower indexes in premalignant breast lesions [18,23,58]. This is in line with our results, where Ki-67 index presented a mean of 12.8%, and 37 lesions (37/47, 78.7%) had a low Ki-67 index. Ki-67 index has a prognostic value in feline mammary carcinoma and is higher in more aggressive tumors [39]. To the best of our knowledge, only one study evaluated Ki-67 index in benign feline lesions. However, they found the lowest index in dysplastic lesions (mean 6.9%) followed by benign feline tumors (mean 7.5%) [48], while in the present study, we found that benign tumors presented lower Ki-67 values (10.6%), followed by the dysplastic lesions (13.2%). Additionally, we did not find any significant association between Ki-67 and other analyzed biomarkers or clinicopathological features.

Finally, the expression of cytokeratins 5/6, which is used as an indicator of more aggressive types of neoplasias, was also evaluated. Although CK 5/6 expression is routinely used in HBC to determine or confirm histopathologic diagnosis [59], its expression in benign feline mammary lesions has not yet been investigated. In our study, most of the lesions were considered negative for CK 5/6 immunostaining, confirming their benign nature. 

## 5. Conclusions

The results here presented uncover novel findings regarding benign feline mammary (neoplastic and non-neoplastic) lesions and complement previously published results, adding relevant information to the state of the art. Indeed, the study of the initial lesions is essential in order to understand the progression of the disease and to develop better prophylactic strategies, with the ultimate goal to prevent mammary lesion progression.

## Figures and Tables

**Figure 1 animals-12-01599-f001:**
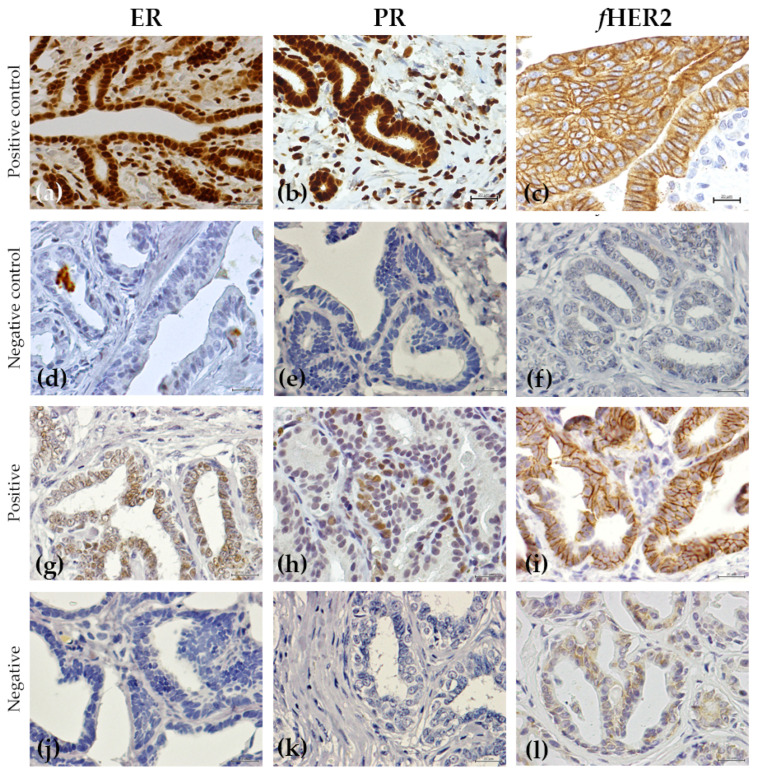
Immunohistochemical expression of the different proteins studied in non-neoplastic benign tumors and in benign tumors: (**a**–**c**) positive controls for ER, PR and *f*HER2, respectively; (**d**–**f**), negative controls for ER, PR and *f*HER2, respectively; (**g**) fibroadenomatous change with a positive score for ER (Allred score 7/8); (**h**) simple adenoma with a positive score for PR (Allred score 4/8); (**i**) simple adenoma classified as positive for *f*HER2 (3+) and (**j**) as negative for ER; (**k**) fibroadenomatous change with a negative score for PR and; (**l**) a simple adenoma classified as negative for *f*HER2 (0). All IHC were counterstained with Mayer’s hematoxylin (400×).

**Figure 2 animals-12-01599-f002:**
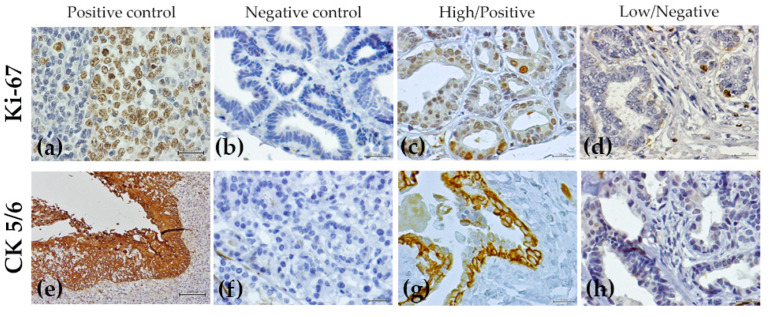
Immunohistochemical expression of the Ki-67 and CK5/6 proteins in non-neoplastic benign tumors and in benign tumors: (**a**) Ki-67 positive control, feline tonsil; (**b**) negative control for Ki-67; (**c**) simple adenoma presenting a high Ki-67 proliferation index (29%); (**d**) fibroadenomatous change with low Ki-67 index (1%); (**e**) positive control for CK5/6, oral epithelium (100×); (**f**) CK 5/6 negative control; (**g**) ductal ectasia presenting a positive staining for CK5/6 and; (**h**) a simple adenoma with a negative score for CK5/6. All the samples were counterstained with Mayer’s hematoxylin (400×).

**Table 1 animals-12-01599-t001:** IHC classification criteria for HER2, ER and PR evaluation.

**HER2 Score**		
0	No staining or membrane staining that is incomplete, weak and in ≤10% of lesion cells.	
1+	Incomplete membrane staining that is weak in >10% of lesion cells.	
2+	Weak to moderate complete membrane staining observed in >10% of lesion cells.	
3+	Circumferential membrane staining that is complete, intense and in >10% of lesion cells.	
**ER/PR Score**		
Negative	Nuclear staining in <1% of lesions cells.	
Positive	Nuclear staining in ≥1% of lesions cells.	

**Table 2 animals-12-01599-t002:** Allred score guidelines for ER and PR evaluation.

**Score for the percentage of positive lesion cells**	
0	No staining.
1	<1% of nuclear staining.
2	1–10% of nuclear staining.
3	10–33% of nuclear staining.
4	33–66% of nuclear staining.
5	>66% of nuclear staining.
**Score for average intensity of staining**	
0	None
1	Weak
2	Average
3	Strong
**Allred score = summation of both scores (0–8)**	

**Table 3 animals-12-01599-t003:** Clinicopathologic findings of the 27 cats enrolled in the study.

Features	Number of Animals (%)
**Breed**	
European Shorthair	20 (74.1%)
Persian	4 (14.8%)
Siamese	2 (7.4%)
Norwegian Forest Cat	1 (3.7%)
**Spayed**	
Yes	8 (29.6%)
No	19 (70.4%)
**Contraceptive use**	
Yes	15 (57.7%)
No Information not available	11 (42.3%) 1
**Presence of concomitant malignant tumors**	
Yes	16 (59.3%)
No	11 (40.7%)

**Table 4 animals-12-01599-t004:** Histopathologic findings of the 47 lesions analyzed.

Histopathological Group	Histopathological Classification	n (%)	Lesion Size Mean ± SEM (cm)
Mammary hyperplasia and dysplasia		41 (87.2%)	1.77 ± 0.37
	Duct ectasia	23 (48.9%)	1.44 ± 0.46 ^a^
	Fibroadenomatous change	8 (17%)	2.79 ± 0.68 ^b^
	Epitheliosis	4 (8.5%)	0.25 ± 0.9 ^a^
	Lobular hyperplasia (adenosis)	6 (12.7%)	1.22 ± 0.52 ^ab^
Benign neoplasia			
	Simple adenoma	6 (12.8%)	1.75 ± 1.08 ^ab^

^a,b^: Lines with different letters differ significantly (*p* < 0.05). The measurement of the cystic hyperplasia was not considered for this analysis.

**Table 5 animals-12-01599-t005:** Immunohistochemical results for ER, PR, HER2, Ki-67 and CK5/6 staining.

Protein	Total (%)	Benign Non-Neoplastic Lesions (%)	Benign Tumors (%)
**ER status**			
Positive	43 (91.5%)	39 (95.1%)	4 (66.7%)
Negative	4 (8.5%)	2 (4.9%)	2 (33.3%)
**PR status**			
Positive	17 (36.2%)	16 (39%)	1 (16.7%)
Negative	30 (63.8%)	25 (61%)	5 (83.3%)
***f*HER2 status**			
Positive	16 (35.6%)	14 (35.9%)	2 (33.3%)
Negative	29 (64.4%)	25 (64.1%)	4 (66.7%)
Undetermined *	2	2	0
**Ki-67 index**			
mean (max-min)	12.9% (0–52%)	13.2% (0–52%)	10.6% (4–29%)
High	10 (21.3%)	9 (30%)	1 (16.7%)
Low	37 (78.7%)	32 (70%)	5 (83.3%)
**CK5/6 status**			
Positive	11 (23.4%)	9 (30%)	2 (33.3%)
Negative	36 (76.6%)	32 (70%)	4 (66.7%)

* Two tissue samples were lost during the IHC technique determining *f*HER2 status classification.

## Data Availability

The data presented in this study are available on request from the corresponding author in response to reasonable requests.

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
