# Peer review of "fHER2, PR, ER, Ki-67 and Cytokeratin 5/6 Expression in Benign Feline Mammary Lesions"

_animals, 2022, doi:10.3390/ani12131599_

Round 1

Reviewer 1 Report

In the present study, Soares et al. investigate the expression of several immunohistochemical markers, known to be relevant in human breast cancer, in benign lesions of the feline mammary gland. The study is straightforward and with a decent number of cases compared to previous studies available in the literature.

However, some minor revisions are required:

- Throughout the manuscript, the authors refer to benign lesions other than adenomas as “dysplastic lesions”. Dysplasia is a specific terminology implying altered growth, but some of the lesions analyzed here are just hyperplastic ones. I would avoid using the term dysplasia to group all these lesions together. Please, change the terminology (check also the WHO and CL Davis-Thompson foundation classification, where these lesions are grouped under “hyperplastic and dysplastic”).

- Line 107: as not all the lesions included are actually true tumoral lesions change “Tumor” with “Sample”

- Line 119-120: to classify the lesions, the authors referred to the WHO classification (1999). I would suggest adding the more recent classification system published by CL Davis and see if some diagnoses need to be reviewed. (Reference: Zappulli V, Pena L, Rasotto R, et al. Mammary tumors. In: Kiupel M, ed. Surgical Pathology of Tumors of Domestic Animals. Vol 2. Davis Foundation; 2019.)

- Line 121: specify if the measurement of the lesions was performed prior to or after fixation

- Line 126-127: it is unclear if the authors used a tissue microarray technique for this study; if yes this need to be clearly specified.

- Line 158: the authors mention the Allred score. A brief description of the method in the M&M section is warranted. In addition, the results from the Allred score are not reported in the results section (even if this is cited in line 236).

- Line 187: specify if “number of lesions” is referred just to benign lesions or all the mammary lesions (including carcinomas).

- Line 198: there is a discrepancy between this sentence and table 3, from which the second most common lesion seems to be fibroadenomatous hyperplasia (n=8) rather than adenoma (n=6).

- The reviewer suggests to improve the result section including some correlations between the expression of the markers (among themselves). It is unclear if the authors performed this kind of analysis (this is mentioned in the statistical analysis section but not reported in the results).

- Figure 1 and figure 2: some pictures show negative markers. I would suggest to include an inset with the positive control.

- Line 272: not all the lesions included in the study are strictly “tumors”. Change with lesions (and check the whole manuscript for consistency).

- Line 280-281: these data are not reported in the result section. Please, improve the result section or remove them from the discussion.

- Lines 294-296: the sentence “which are usually present in younger animals” need to be supported by a reference. In addition, the correlation between ER expression and younger age does not directly demonstrate a correlation with less aggressive lesions. The sentence needs to be rephrased.

- Lines 304-312: the reviewer has some concerns regarding the biological significance of the correlation between marker expression and size of the lesions, since some of these lesions are cystic (e.g. duct ectasia). For these types of lesions, the size is dependent on the dilation of the duct more than proliferation and thus it is not clear how this might be directly related to the biomarkers.

- 318-320: the sentence is unclear.

- The final part of the discussion/conclusion needs to be improved. It is unclear the overall take-home message of the study. Because not all the benign lesions are necessarily pre-malignant, the authors should discuss if there is any evidence that the described benign lesions are pre-cancerous? If yes, was there any correlation with the markers, and/or do the results support this data? If not, does the results from this study give any insight in this lacking piece of knowledge?

- Some sentences are incorrect and/or need to be rephrased (for example, but not limited to, line 47, line 65, line 85, 97,105, 108, 119, 197….). In general, English revision by a native speaker or a professional editing service is strongly suggested.

Author Response

Reviewer #1 (Comments to the Author):

In the present study, Soares et al. investigate the expression of several immunohistochemical markers, known to be relevant in human breast cancer, in benign lesions of the feline mammary gland. The study is straightforward and with a decent number of cases compared to previous studies available in the literature. Dear reviewer, thank you very much for your words about this manuscript, insightful comments, and constructive remarks along with the document. They really improve the final quality of this article. 

However, some minor revisions are required:

- Throughout the manuscript, the authors refer to benign lesions other than adenomas as “dysplastic lesions”. Dysplasia is a specific terminology implying altered growth, but some of the lesions analyzed here are just hyperplastic ones. I would avoid using the term dysplasia to group all these lesions together. Please, change the terminology (check also the WHO and CL Davis-Thompson foundation classification, where these lesions are grouped under “hyperplastic and dysplastic”).

We have corrected this error throughout the manuscript.

- Line 107: as not all the lesions included are actually true tumoral lesions change “Tumor” with “Sample”

Done.

- Line 119-120: to classify the lesions, the authors referred to the WHO classification (1999). I would suggest adding the more recent classification system published by CL Davis and see if some diagnoses need to be reviewed. (Reference: Zappulli V, Pena L, Rasotto R, et al. Mammary tumors. In: Kiupel M, ed. Surgical Pathology of Tumors of Domestic Animals. Vol 2. Davis Foundation; 2019.)

The lesions were reviewed according to the more recent classification system and some were reclassified (Table 4, line 235).

- Line 121: specify if the measurement of the lesions was performed prior to or after fixation.

Specified in the text.

- Line 126-127: it is unclear if the authors used a tissue microarray technique for this study; if yes this need to be clearly specified.

Yes, we have used the tissue microarray technique and it is clarified in the text (line 133).

- Line 158: the authors mention the Allred score. A brief description of the method in the M&M section is warranted.

A new table was added to describe the classification (Table 2), line 176.

In addition, the results from the Allred score are not reported in the results section (even if this is cited in line 236).

Thank you for the correction, we introduced the allred results in the result section (line 245-250).

- Line 187: specify if “number of lesions” is referred just to benign lesions or all the mammary lesions (including carcinomas).

The analysis considered all the lesions (benign and malignant when they were present). We clarify it in the text (line 199-200).

- Line 198: there is a discrepancy between this sentence and table 3, from which the second most common lesion seems to be fibroadenomatous hyperplasia (n=8) rather than adenoma (n=6).

You are right, it was a mistake, we already correct changing to “the third” (line 217).

- The reviewer suggests to improve the result section including some correlations between the expression of the markers (among themselves). It is unclear if the authors performed this kind of analysis (this is mentioned in the statistical analysis section but not reported in the results).

Yes, we have done it, but no significant results were found. We added this information on lines 182-183 and 281-282 of the manuscript.

- Figure 1 and figure 2: some pictures show negative markers. I would suggest to include an inset with the positive control.

As suggested, positive controls were inserted.

- Line 272: not all the lesions included in the study are strictly “tumors”. Change with lesions (and check the whole manuscript for consistency).

Done throughout the manuscript.

- Line 280-281: these data are not reported in the result section. Please, improve the result section or remove them from the discussion.

As required, more information was added to the results’ section, in order to further support the discussion (line 299-312).

- Lines 294-296: the sentence “which are usually present in younger animals” need to be supported by a reference.

In cats, the available literature indicates age as a risk factor, with older queens presenting more probability to develop malignant mammary tumors (Hayden, D.; Nielsen, S. (1971) Feline mammary tumours. Journal of Small Animal Practice 12 (12): 687-698; Hoepp, N (2021): Mammary Gland. in Sharkey,L ; Radin, M; Seelig, D; Veterinary Cytology, 1ª Edição, Pp 582-593 , USA, John Wiley & Sons).

However, there are few references that clearly say that in the younger animals the lesion will be benign or non-neoplastic (Graf, R., Grüntzig, K., Hässig, M., Axhausen, K. W., Fabrikant, S., Welle, M., … Pospischil, A. https://doi.org/10.1016/j.jcpa.2015.08.007), which lead us to delete that information from the manuscript.

In addition, the correlation between ER expression and younger age does not directly demonstrate a correlation with less aggressive lesions. The sentence needs to be rephrased.

We rephrased the sentence in an effort to be more explicit (line 372).

- Lines 304-312: the reviewer has some concerns regarding the biological significance of the correlation between marker expression and size of the lesions, since some of these lesions are cystic (e.g. duct ectasia). For these types of lesions, the size is dependent on the dilation of the duct more than proliferation and thus it is not clear how this might be directly related to the biomarkers.

Thank you for this very important comment. We have performed the statistical analysis removing the ductal ectasias and it was still significant (p=0,038). We have altered the text, accordingly (line 276-280).

- 318-320: the sentence is unclear.

We have reformulated the sentence, to become more understandable (line 396-400).

- The final part of the discussion/conclusion needs to be improved. It is unclear the overall take-home message of the study. Because not all the benign lesions are necessarily pre-malignant, the authors should discuss if there is any evidence that the described benign lesions are pre-cancerous? If yes, was there any correlation with the markers, and/or do the results support this data? If not, does the results from this study give any insight in this lacking piece of knowledge?

As suggested, the discussion was improved. Being a descriptive study, our results aim to characterize the lesions and point out the importance of studying them. Unfortunately, we were not able to get a high number of significant associations, but we think that is important to describe the molecular profile of the non-neoplastic mammary lesions and benign mammary tumors to prompt more research that could potentially clarify some of the doubts pointed out here.

- Some sentences are incorrect and/or need to be rephrased (for example, but not limited to, line 47, line 65, line 85, 97,105, 108, 119, 197….). In general, English revision by a native speaker or a professional editing service is strongly suggested.

Dear reviewer, thank you for this recommendation. The article was reviewed by a native English speaker.

Reviewer 2 Report

Dear authors,

This work analyses the expression of different molecules related with tumour progression in mammary benign lesions in cats. The results are very interesting and demonstrate that oestrogen receptors have a relevant role in feline tumours progression. The introduction provides sufficient background and includes relevant references, the cited references are relevant to the research, the research design is appropriate, the methods are adequately described, the results are clearly presented, and the conclusions are supported by the results. Only, one change is necessary before the publication of this study. In the 203 line, the authors indicate “were significantly bigger”. Given that a result is significant or not, I recommend that authors remove "bigger" from this sentence.

I would like to congratulate the authors, since I consider it to be a very interesting study from the point of view of the veterinary clinic.

1.       What is the main question addressed by the research? 

The main question addressed by the research is the study of the differential expression of estrogen and progesterone receptors in mammary tumors in cats. 

2.       Do you consider the topic original or relevant in the field, and if so, why? 

This topic is very original, so some studies about the expression of receptors estrogens and progesterone related to progression of tumours in cats, do not exist. 

3.       What does it add to the subject area compared with other published material? 

Although there are some studies where the expression of these receptors in other tumors is analyzed, there is only one published work in the literature where estrogen receptors are analyzed in breast tumors, and none where estrogen and progesterone receptors are analyzed simultaneously. 

4.       What specific improvements could the authors consider regarding the methodology? 

I consider the methodology is correct. 

5.       Are the conclusions consistent with the evidence and arguments presented and do they address the main question posed? 

The conclusions are consistent with the evidence and arguments presented. 

6.       Are the references appropriate? 

The references are appropriate. 

7.       Please include any additional comments on the tables and figures. 

The tables and figures on the manuscript are clear and correct for its interpretation.

Author Response

Dear reviewer 2,

Thank you so much for your positive words about our work. Your comments generate a great stimulus to go forward with our mission. We much appreciate that. As suggested, the word "bigger" in line 203 was removed.

Reviewer 3 Report

The manuscript entitled “fHER2, PR, ER, Ki-67 and cytokeratin 5/6 expression in feline 2 mammary benign lesions” described the expression of several markers on feline benign lesions of mammary gland. This topic in poorly explored and represents an interesting contribution. Please, see my specific comments below:

1.       The abstract was poorly explored and is too superficial. There is no mention regarding the benign lesions diagnosis. Authors could explore more the diagnosis of each benign lesion. How many cats showed more than one diagnosis?

2.       Methods section. Benign mammary lesions mean benign tumors? Because paraneoplastic lesion is also benign lesions. However, authors claimed to perform surgery due tumors in “animal population subheading”. Therefore, authors could check out carefully the nomenclature.

3.       Subheading “Mammary tissue collection and histopathological classification”. It is unclear the sample population. Authors mentioned that patients had malignant tumors. Therefore, it is not understandable how authors received the samples.

4.       A serious flaw, authors mentioned positive control but not negative control for IHC.

5.       Only in results (later in manuscript) it is possible to understand that benign non-neoplastic lesions and benign tumors were both included in this manuscript.

6.       Why the malignant tumors were no included?

7.       Results section, phrase “Overall lesion mean size at the time of surgery was 1,76 cm (SEM ±0,35), ranging 200 between 0,1 and 10 cm.” mean size of what? Benign tumors? Because the benign non-neoplastic lesion were not macroscopic. Is that right?

8.       The figure 1, image c is a negative HER image. How about a positive one?

9.       Authors could explore more the figure including image of positive and negative controls, and a positive and negative image from each marker.

Author Response

Reviewer #3 (Comments to the Author):

The manuscript entitled “fHER2, PR, ER, Ki-67 and cytokeratin 5/6 expression in feline 2 mammary benign lesions” described the expression of several markers on feline benign lesions of mammary gland. This topic in poorly explored and represents an interesting contribution. Please, see my specific comments below:

  1. The abstract was poorly explored and is too superficial. There is no mention regarding the benign lesions diagnosis. Authors could explore more the diagnosis of each benign lesion. How many cats showed more than one diagnosis?

Dear reviewer, thank you for this suggestion. The abstract was improved and the different diagnosis in the female cats were more detailed in the text (line 208-212). Regarding, the animals with multiple diagnosis:  16 animals showed malignant tumours simultaneously. From the 27 cats, 12 presented more than 1 benign lesion (neoplastic or non-neoplastic) at the time of the diagnosis. From these 12 cats, 5 cats presented different benign lesions (e.g. one dysplastic and one hyperplastic lesion in different mammary glands).

  1. Methods section. Benign mammary lesions mean benign tumors? Because paraneoplastic lesion is also benign lesions. However, authors claimed to perform surgery due tumors in “animal population subheading”. Therefore, authors could check out carefully the nomenclature.

The text was improved and clarified (line 110).

  1. Subheading “Mammary tissue collection and histopathological classification”. It is unclear the sample population. Authors mentioned that patients had malignant tumors. Therefore, it is not understandable how authors received the samples.

The sample population were benign lesions (benign tumors and non-neoplastic benign lesions like hyperplasia and dysplasia lesions). Some of the animals that presented these lesions, also presented, at the same time but in other(s) mammary gland(s), malignant lesions. However, these malignant lesions were not used for the immunohistochemistry analysis and were not included in the sample population. We only used the information if the animal presented or does not present, simultaneously, malignant mammary lesions for statistical studies, but no significant associations were found. The text was improved to be more clear (line 126-127).

  1. A serious flaw, authors mentioned positive control but not negative control for IHC.

Dear reviewer, thank you for this important suggestion. A negative control was added to the text (line 157).

  1. Only in results (later in manuscript) it is possible to understand that benign non-neoplastic lesions and benign tumors were both included in this manuscript.

We have clarified it better in the material and methods section (line 110).

  1. Why the malignant tumors were no included?

Because the main objective of our study was to focus on the non-malignant lesions. All the articles that we have found regarding this topic in veterinary studies refer the malignant lesions, which ends up making the others secondary and poorly studied and described.

  1. Results section, phrase “Overall lesion mean size at the time of surgery was 1,76 cm (SEM ±0,35), ranging between 0,1 and 10 cm.” mean size of what? Benign tumors? Because the benign non-neoplastic lesion were not macroscopic. Is that right?

Dear reviewer, it was the mean size of all the lesions that was evaluated. We have clarified the sentence to be easier to understand (line 219).

  1. The figure 1, image c is a negative HER image. How about a positive one?

We have added an image with a HER2 positive lesion (Figure 1, line 287).

  1. Authors could explore more the figure including image of positive and negative controls, and a positive and negative image from each marker.

Dear reviewer, thank you for this suggestion. As requested, we have added images from positive and negative controls and positive and negative lesions (Figure 1, line 287 and Figure 2, line 313).

Round 2

Reviewer 3 Report

I have no further comments.